# Onion pollenkitt: Function, DNase activity, fatty acid composition, and SEM imaging

**Majd Mardini**[1,2], **Aleksey Ermolaev**[1,2], **Alexey Yu. Kuznetsov**[1], **Alexey V. Zhevnerov**[1], **Sergey Buravkov**[3,4], **Ludmila Khrustaleva**[1,2]*

1 Center of Molecular Biotechnology, Russian State Agrarian University—Moscow Timiryazev Agricultural Academy, Moscow, Russia, 2 Laboratory of Marker-Assisted and Genomic Selection of Plants, All-Russian Research Institute of Agricultural Biotechnology, Moscow, Russia, 3 Faculty of Fundamental Medicine, M. V. Lomonosov Moscow State University, Moscow, Russia, 4 Institute for Biomedical Problems, Russian Academy of Sciences, Moscow, Russia

* ludmila.khrustaleva19@gmail.com

## Abstract

Pollenkitt, a sticky substance produced by the tapetal cells of pollen grains, plays a crucial role in pollen functionality, yet it remains an understudied component in plant biology. In this study, we investigated the role of pollenkitt in onion (*Allium cepa*) pollen, focusing on its effects on pollen germination, DNase activity, fatty acid composition, and ultrastructure. Our findings reveal that pollenkitt is essential for successful onion pollen germination and tube growth on the stigma, as its removal significantly inhibited these processes. Additionally, we demonstrated that onion pollenkitt exhibits DNase activity, degrading plasmid DNA in a concentration-dependent manner, with EDTA effectively inhibiting the degradation. Gas chromatography identified 20 fatty acids in pollenkitt, with a predominance of unsaturated fatty acids. Using scanning electron microscopy (SEM), we showed structural differences between pollen grains with and without pollenkitt, and we observed the process of pollenkitt removal from the surface of pollen grains by water washing. These results offer valuable insights for future research aimed at optimizing pollen-mediated gene-editing systems and highlight the importance of considering pollenkitt in such applications.

## Introduction

In cross-pollinating plants, pollen grains play a crucial role as carriers of the male gametes in order to be transported to their female counterparts. After meiosis, the developing pollen grains, known as microspores, are released from the tetrad stage while being encased in a special cell wall [1]. A matrix of polysaccharides arises from this special wall, creating a structure known as primexine [2–4]. At this point, a complex substance called sporopollenin is synthesized in the tapetal cells and accumulates in the locules of the anthers. Sporopollenin is then deposited onto the primexine and later undergoes polymerization, leading to the formation of the outer layer of the pollen wall, known as the exine [5]. In addition to sporopollenin, the tapetal cells are also responsible for producing other pollen coating materials such as pollenkitt and tryphine [6]. Pollenkitt is relatively more prevalent than tryphine (tryphine is limited only to Brassicaceae) and is primarily characteristic of most entomophilous and zoophilous species [6–9]. After synthesizing pollenkitt the tapetum cells initiate programmed death [10].

**Data availability statement:** All relevant data are within the manuscript and its Supporting Information files.

**Funding:** This research was supported by the Ministry of Science and Higher Education of the Russian Federation, agreement No. 075-152022-317 dated 20 April 2022, "Agrobotechnologies of the Future". SEM analysis was partly supported by the Russian Foundation for Basic Research, grant number 20-016-00065.

**Competing interests:** The authors have declared that no competing interests exist.

Eventually, the tapetal cells disintegrate, leading to the complete break-down of the tapetum and the release of cellular remnants, including pollenkitt, which envelop the fully developed pollen grains [11]. The term Pollenkitt was first coined by Knoll in 1930 [7], originally used to describe all sticky substances that combine pollen grains into clumps.

Pollenkitt is composed mainly of lipids, as well as small amounts of carotenoids, flavonoids, proteins and carbohydrates [6,12,13]. Pollenkitt's lipids contain a large amount of fatty acids, which are essential for proper pollen–stigma communication [14] and attracting pollinators [15]. To date, the quantitative chemical composition of fatty acids in pollenkitt has been characterized for several plant species including *Brassica napus* [16], sunflower [17], clover [18], maize [19], *Crocus vernus* [20], spruce, tobacco and lily [21]. Despite the limited information available, it is already apparent that there is significant variation in the chemical composition of pollenkitt fatty acids in different species and even in different varieties [19]. Along with several core fatty acids, which are essential for normal pollen functioning, various derivatives could be found.

The pollenkitt fulfills important role in life cycle of mature pollen. Hesse [6] re-viewed twenty pollenkitt functions at different stages of the pollen grain life cycle, including dispersal, capture by pollinators, attachment to stigma, and germination. However, given the multifunctional properties of pollenkitt, it remains one of the least studied objects in plant biology and biotechnology. A search for "pollenkitt" on Google Scholar shows about 700 published works over the past five years, which is strikingly low compared to other related keywords. For instance, in the same timeframe, there are 50,000 related publications for the keyword "pollination" and 18,000 for "pollinators". Even more specialized terms, such as "sporopollenin" and "tapetum", yield substantially higher numbers of publications, with around 4,700 and 8,400, respectively. This simple comparison highlights the limited research focus on pollenkitt compared to other key topics in the field, not to mention the scarcity of studies on the pollenkitt of agriculturally important species. There has been a recent growing interest in using pollen grains as natural carriers for gene-editing systems targeting the generative cells of plants [22–27]. However, pollenkitt is totally over-looked in this field of research.

In this study, we investigated different aspects of the pollenkitt of the species *Allium cepa*. Bulb onion is an important vegetable crop, ranking second after tomato ac-cording to a FAO report (https://www.fao.org/3/cc3751en/cc3751en.pdf - accessed on 26.11.2024). Many studies have been conducted to explore various areas in the field of onion palynology, including issues such as male sterility [28,29], seed production [30], pollination and bees [31], fundamental biology [32], and others. However, research on pollenkitt in this species remains scarce. Prior to our recent study [33], no significant research had been conducted on onion pollenkitt. To our knowledge, this was the first work to explore various critical aspects of onion pollenkitt, including the molecular analysis of pollen grains, wet pollination, and structural examination using light microscopy.

Here, we further explore key aspects of onion pollenkitt: 1) the critical role of pollenkitt in pollen tube germination *in vivo*; 2) we provided evidence of the DNase activity of the onion pollenkitt; 3) we determined the fatty acid composition of onion pollenkitt using gas chromatography; 4) we performed scanning electron microscopy (SEM) of onion pollen grains with and without pollenkitt.

## Materials and methods

### Plant material

*Allium cepa* cv. 'Myachkovsky 300' was grown on the experimental field of the Molecular Biotechnology Center, Russian State Agrarian University – Moscow Timiryazev Agricultural

Academy. By late April, vernalized onion bulbs were planted with a spacing of 40 cm. At the height of the flowering period (late July), umbels with 60–70% of their flowers open were cut and placed in a beaker filled with tap water. A total of 7 umbels from 7 plants were incubated at 24 °C throughout the pollen collection period. Since flowers within a single umbel do not develop simultaneously, pollen was collected once or twice daily over 3–4 days. By gently tapping the umbel over a glass Petri dish, pollen grains from open anthers were dispersed across the glass. Clumps of pollen were scraped from the surface of the glass using a pair of razor blades. Pollen from the umbels were bulked into a single Petri dish (S1 Fig). The collected pollen was either used immediately or stored at -80 °C.

## Onion pollen germination and tube growth *in vivo*

**Pollenkitt removal and manual pollination.** Three different variants of manual pollination were performed: (1) pollination with dry pollen (control); (2) pollination with pollen suspension without pollenkitt removal; (3) pollination with pollen suspension after pollenkitt removal. For all three variants, fresh mature pollen grains were collected from 7 umbels from 7 plants, as shown in section 4.1. Dry pollen used in variant (1) was applied to the stigma using a paintbrush. Pollen suspensions in variants (2) and (3) were applied to the stigma using a microbiological loop. The preparation of these suspensions was conducted as follows: Using a sensitive analytical balance (Ohaus-PA214C), 1 mg of mature pollen grains was weighed in a 1.5 mL tube. To create a uniform suspension, 100 μL of a 12% sucrose water solution was added to the tube, and the pollen grains were thoroughly mixed using a pipette tip. The resulting suspension was then divided into two clean tubes, each containing 50 μL. These tubes underwent vortexing for 10 seconds, followed by 5 seconds of centrifugation at 750 g. This procedure facilitated the detachment of pollenkitt from the pollen grains' surface into the supernatant, while the pollen grains precipitated to the bottom of the tube. Next, in one tube, the supernatant was retained, and the precipitated pollen grains were re-suspended in it (variant 2). In the other tube, the supernatant was discarded, and the precipitated pollen grains were re-suspended in fresh 50 μL of 12% sucrose (variant 3).

In all the variants, manual pollination was performed only when the stigmatic knob was visible, indicating optimal stigma receptivity for pollen grains [34] (see S2 Fig). Each of the three variant was repeated on five different flowers from five different umbels (i.e., 15 pollinations were carried out in total). Pollination was performed in field conditions. The pollinated umbels were isolated from insects using cages made from medical gauze.

**Aniline blue staining and fluorescent microscopy.** Staining with aniline blue of onion styles was carried out following the protocol by Mori [35], with some modifications. In summary, 16 hours after pollination, flowers were collected using forceps. After removing the tepals, the remaining part of the flower (style and ovary attached to the stem) was fixed in a solution of 3:1 ethanol to acetic acid for 2 hours at room temperature. The fixative was gradually exchanged by incubating in 70%, 50%, 30% ethanol, and distilled water for 10 minutes each. The specimens were later softened by incubating in 12 M NaOH for 4 hours. After that, specimens were carefully washed with distilled water for a couple of times. Staining was performed at room temperature for 2 hours using decolorized 0.1% (w/v) aniline blue solution in 108 mM K3PO4 (pH = 11). After staining, the style was carefully detached from the ovary using a sharp scalpel and forceps, then mounted on a microscope slide in a drop of 100% glycerol. A coverslip was gently applied to flatten the specimen. Visualization was carried out using a Zeiss Axio Imager.M2 fluorescent microscope equipped with an EC Plan-Neofluar 10x/0.30 objective and a DAPI filter (excitation wavelength 359–371 nm). Imaging was conducted with a Hamamatsu camera, utilizing the ZEN-Blue software and the panorama method.

### Investigating DNase activity in onion pollenkitt

To assess whether the pollenkitt from onion pollen exhibits DNase activity, we conducted a simple and straightforward experiment: a fixed amount of plasmid DNA was incubated at room temperature in a pollenkitt suspension, with variations in both incubation time and pollenkitt concentration. As demonstrated in our previous work [33], onion pollenkitt can be easily collected from mature pollen grains by vortexing in an aqueous solution. In this study, we followed the same approach to collect onion pollenkitt. Special care was taken to roughly estimate the mass of pollenkitt used during the incubation with plasmid DNA. The experimental procedure was carried out as follows:

### Preparation of pollenkitt suspensions at various concentrations

Using a sensitive analytical balance (Ohaus PA214C), we weighed 10 mg of freshly collected mature pollen grains into a 2 mL microtube. We then added 1 mL of $dH_2O$ (distilled water) to the tube and vortexed it for 10 seconds. To separate the pollen grains, we performed a short 5-second spin in a benchtop centrifuge at 750 g. The lighter pollenkitt particles remained in the cloudy supernatant, which we transferred to a clean microtube. The supernatant was further centrifuged at 12,000 g for 1 minute to precipitate the pollenkitt particles. After centrifugation, the pollenkitt formed a pale yellow pellet at the bottom of the tube. We carefully removed the now-clear supernatant with a pipette, leaving the pellet behind. The pollenkitt pellet was dried at room temperature for 1 hour, resulting in a final mass of approximately 1.7 mg. We re-suspended the dried pollenkitt in 1700 µL of $dH_2O$ to create a suspension of pollenkitt with a concentration of 1 µg/µL. By diluting 1, 2 and 5 µL of this suspension with $dH_2O$ to a final volume of 10 µL, we created different concentrations of pollenkitt suspension: 0.1 µg/µL, 0.2 µg/µL and 0.5 µg/µL, respectively. Each of the suspensions, including 1 µg/µL, was prepared in six replicates. In each replicate, a fixed amount of plasmid DNA was added, and the mixture was incubated at room temperature for six different durations, as described in the next section.

### Incubation of plasmid DNA in various concentrations of pollenkitt suspension over different time durations

The plasmid DNA we used was from Addgene (Addgene plasmid #80129; http://n2t.net/addgene:80129 – accessed 01.10.2024). The total size of this plasmid is 4810 bp. The plasmid DNA was diluted in double-distilled water to a working concentration of 100 ng/µL. The experiment involved incubating each pollenkitt concentration (0.1 µg/µL, 0.2 µg/µL, 0.5 µg/µL and 1 µg/µL) with plasmid DNA for six different time intervals (10 minutes, 15 minutes, 30 minutes, 60 minutes, 90 minutes, and 120 minutes). 1 µL of plasmid DNA was added to 9 µL of each pollenkitt concentration. All 24 incubation reactions (4 different pollenkitt concentrations × 6 different durations) were run on a 1% agarose gel at 120 volts for 45 minutes. Two controls were performed: (1) 9 µL $dH_2O$, 1 µL plasmid DNA (100 ng/µL) with six incubation times; (2) 1, 2, and 3 µL of EDTA (0.5 M, pH 8.0) and 8, 7, 6 µL of 1µg/µL pollenkitt, and 1 µL plasmid DNA (100 ng/µL) with 10 minutes of incubation time.

### Gas chromatography with flame ionization detector (GC-FID) of pollenkitt composition

**Lipid extraction and fatty acid methyl esters preparation.** Probe preparation of fatty acids for GC-FID was performed according to ISO 12966-2-2011. A total of 30 mg of pollen was suspended in 1 mL of hexane, a non-polar solvent, using metal beads. The extraction

was performed using TissueLyser II (Qiagen) for 1 minute at 3 Hz. Then, 0.15 ml of a 2 M sodium methylate solution in anhydrous methanol was added to the tube and quickly mixed. After centrifugation at 14,000 RCF for 10 minutes, the upper hexane layer containing fatty acid methyl esters was filtered through a paper filter into a new tube and placed on the chromatograph for analysis.

**Performing gas chromatography and injection conditions.** Fatty acids methyl esters were analyzed using gas chromatography with flame ionization detector. Analysis was performed on a Shimadzu Nexis GC-2030 AF gas chromatograph (Shimadzu, Kyoto, Japan) with a flame ionization detector FID‒2030 fitted with a 30-m capillary column Omegawax 250 (inner diameter 0.25 mm, thickness of stationary phase—0.25 μm). The prepared fatty acids' methyl esters were separated under the following conditions: nitrogen at 1 mL/min as a carrier gas; 1 μL of sample; 10:1 split ratio; 250 °C evaporator temperature. The oven temperature program was as follows: from 60 to 170 °C at 20 °C/min, from 170 °C to 260 °C at 2 °C/min. The operational temperature of the flame ionization detector was set to 260 °C. Identification of individual fatty acid methyl esters was performed by retention time using a standard mix of 37 fatty acids methyl esters Supelco-CRM47885. The unsaturation index was calculated according to the method described by Chen and Liu [36]. Raw data on retention time area, along with the chromatogram, are available in the supplementary materials (S1 Table and S3 Fig).

## SEM analysis of onion pollen grains with and without pollenkitt

The native dry pollen grains were mounted to adhesive double-sided carbon tape and gold coated for SEM analysis.

The pollen grains in water after centrifugation and discarding of the supernatant were air-dried and mounted to adhesive double-sided carbon tape and gold coated.

The pollen grains, after the removal of pollenkitt through 10 rounds of water washing, vortexing, centrifugation, and discarding of the supernatant, as described in our previously published protocol [33], were air-dried and gold-coated for SEM analysis.

SEM imaging of pollen grain exine was conducted following the protocol described in our previous publication [37]. Briefly, fixation in 2.5% formaldehyde following dehydrated by sequential washing with 70%, 90%, and 96% ethanol, soaking in hexamethyldisilazane (HMDS), air drying and gold coating. The images were taken under a scanning electron microscope JEOL JCM-7000.

# Results

## Pollenkitt role in pollen tube germination

In order to evaluate the role of pollenkitt in pollen tube germination *in vivo* onion flowers were manually pollinated using (1) dry pollen, (2) pollen suspension without removal of pollenkitt, and (3) pollen suspension after removal of pollenkitt (refer to section 4.2.1 for more details).

Aniline blue staining showed that pollen tubes in the control variant (1) grew normally as expected (Fig 1a). Similarly, pollen tubes in variant (2) exhibited normal growth along the style after wet pollination with pollen suspension containing pollenkitt (Fig 1b). However, in variant (3) with thoroughly washed off pollenkitt and wet pollination with pollen suspension, no pollen tube or only one was observed, while others were visible near the stigma but did not grow through the entire style (Fig 1c). Notably, the pollen suspensions in variants (2) and (3) underwent identical mechanical preparation, with the sole difference being the presence or absence of pollenkitt. This confirms that the failure of pollen tube germination in variant (3) was due to the absence of pollenkitt rather than the preparation method.

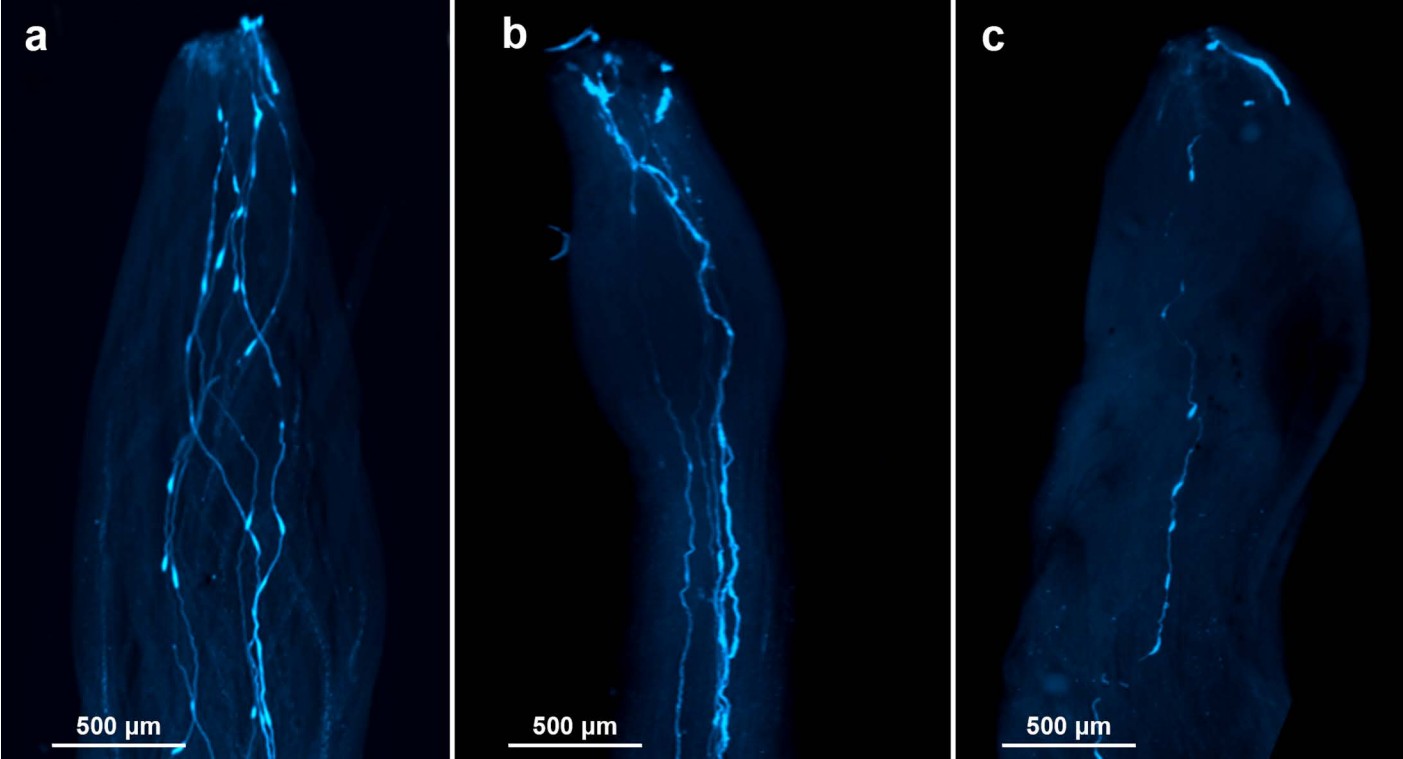

**Fig 1. Fluorescent microscopy of pollen tube germination within the style of the pistil of onion (aniline blue-stained germinated pollen tube). (a)** Pollination with dry pollen grains; (b) pollination with pollen suspension without removal of pollenkitt; (c) pollination with pollen suspension after removal of pollenkitt. Numerous pollen tubes are observed along the style (a,b). Only one pollen tube is observed along the style after the removal of pollenkitt **(c)**.

### Evidence of DNase activity of the pollenkitt of onion

In this part of the study, we aimed to determine whether DNase activity is pre-sent in onion pollenkitt. By incubating plasmid DNA with pollenkitt extracted from mature onion pollen grains, we observed significant degradation of the plasmid DNA. The experiment was designed to estimate the degree of DNA degradation depending on pollenkitt concentration and incubation time.A fixed amount (100 ng) of plasmid DNA in 10 μL of aqueous suspension containing 0.1 μg/μL, 0.2 μg/μL, 0.5 μg/μL, and 1.0 μg/μL of pollenkitt was incubated for 10, 15, 30. 60. 90 and 120 minutes. After the incu-bation, the integrity of plasmid DNA was visualized using gel electrophoresis. It was surprising to observe that DNA degradation occurred after a short incubation for 10 minutes with the lowest concentration (0.1 μg/ μL) of the pollenkitt suspension (Fig 3a). By increasing the concentration of the pollenkitt, the degradation of the plasmid DNA was strongly affected (Fig 2b-d). Furthermore, at concentrations of 0.5 μg/ μL of pollenkitt, the plasmid DNA was completely digested after a short incubation for 10 minutes. An obvious correlation between the level of DNA degradation and the incubation time was observed (Fig 2a-d). At pollenkitt concentrations of 0.5 μg/ μL and 1 μg/ μL, no detectable traces of plasmid DNA were observed following 60 minutes or more of incubation (Figs 2c and d).

To provide additional proof that plasmid degradation was indeed due to enzymatic activity, we conducted an experiment in which EDTA was added to the plasmid-pollenkitt reaction. The chelating properties of EDTA are commonly employed to inhibit enzymatic activity, so its addition to the pollenkitt-plasmid reaction was expected to prevent degradation of the

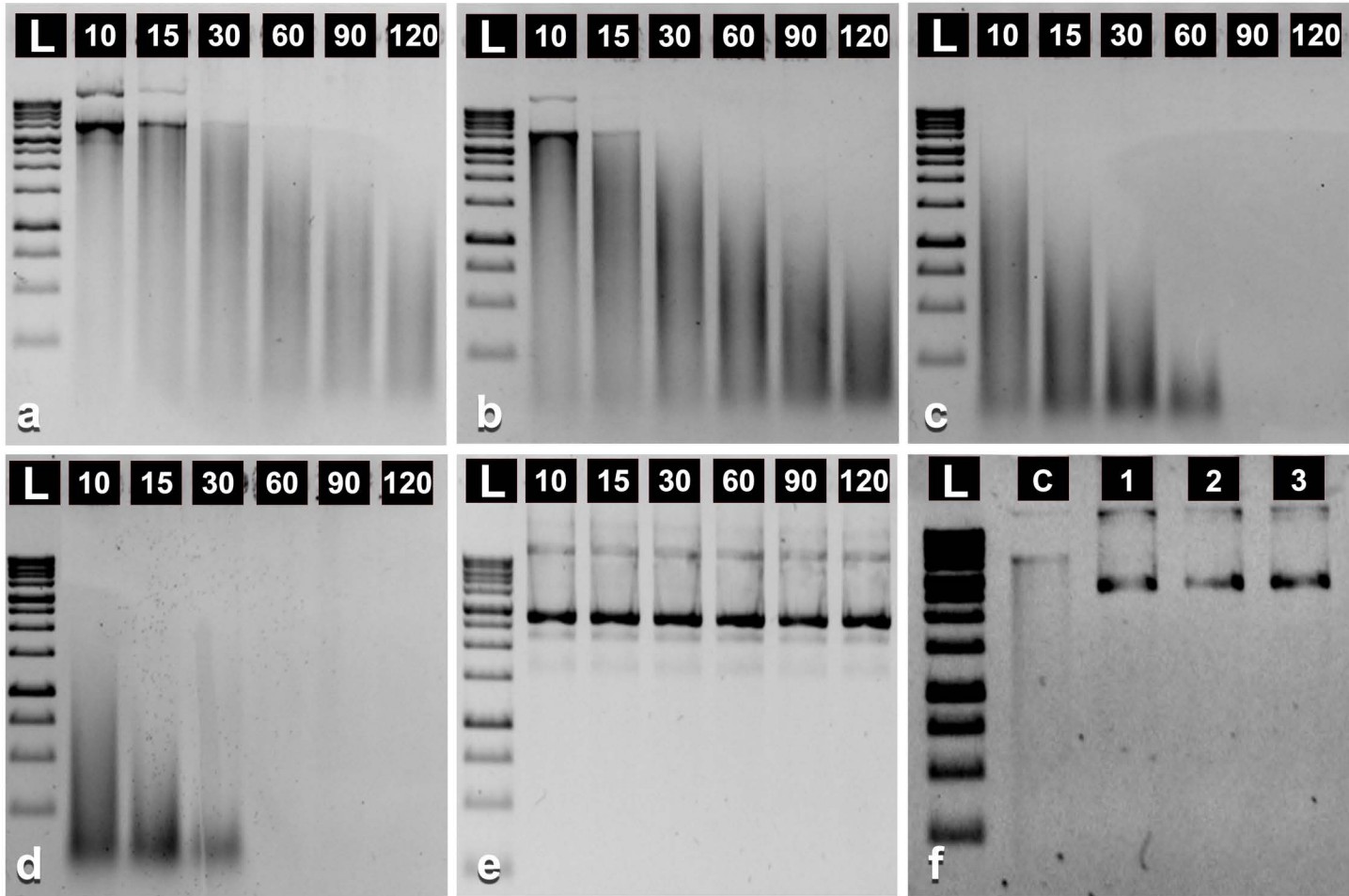

**Fig 2. Gel electrophoresis of 100 ng plasmid DNA following various treatments.** (a) in 10 μL of 0.1 μg/μL aqueous suspension of pollenkitt; (b-d) similar to (a) but with increasing amounts of pollenkitt: 0.2 μg/μL, 0.5 μg/μL, and 1 μg/μL, respectively; **(e)** Control: the plasmid DNA in 10 μL of distilled water without pollenkitt. Numbers above each lane in panels a-e indicate the different incubation times in minutes; (f) addition of EDTA (Ethylenediaminetetraacetic) to the plasmid DNA in a 0.1 μg/μL aqueous suspension of pollenkitt, incubation for 15 minutes. Numbers above each lane indicate the volume of added EDTA (0.5 M) in microliters. **C:** Control, without EDTA. **L:** 1 kb DNA ladder (Evrogen, Russia). Incubation in all variants was carried out at room temperature. All original uncropped gel images can be found in S1 raw images file.

plasmid DNA. Our experiment revealed the anticipated results: even at the minimal concentration, the addition of EDTA effectively halted the DNase activity of the pollenkitt (Fig 2f). This further supports the hypothesis that the degradation of plasmid DNA resulted from the presence of DNase or DNase-like enzyme in the pollenkitt.

### Fatty acid composition of onion pollenkitt

The main components of pollenkitt are fatty acids. The analysis of fatty acid composition was performed using gas chromatography with flame ionization detector (GC-FID).

A total of 20 fatty acids were identified, comprising both saturated and unsaturated types, with side chain lengths varying between 12 and 24 carbon atoms, as presented in Table 1 (a chromatogram is provided in S3 Fig). Unsaturated fatty acids with one (monoene), two (diene), three (triene), and four (tetraene) double bonds were detected. A notable abundance of 16:0 (palmitic), 18:1(9) (oleic) and 18:2(9,12) (linolenic) fatty acids were observed, constituting approximately 18.72%, 16.47% and 11.49%of the total fatty acids of onion pollenkitt,

**Table 1. The percentage of fatty acids in *A. cepa* pollenkitt according to GC-FID.**

| Fatty acid | | % in pollenkitt[*] |
|---|---|---|
| 12:0 | Lauric | 1.33 |
| 13:0 | Tridecylic | 0.88 |
| 14:0 | Myristic | 4.52 |
| 14:1(9) | Myristoleic | 1.04 |
| 15:0 | Pentadecylic | 3.29 |
| 16:0 | Palmitic | 18.72 |
| 17:0 | Margaric | 1.44 |
| 17:1(10) | 10-heptadecaenoic | 5.34 |
| 18:0 | Stearic | 3.93 |
| 18:1(9) | Oleic | 16.47 |
| 18:2(9,12) | Linoleic | 11.94 |
| 18:3(9,12,15) | α-linolenic | 5.05 |
| 20:0 | Arachidic | 1.29 |
| 20:1(9) | *cis*-11-Eicosenoic | 0.85 |
| 20:2(11,14) | Eicosadienoic | 4.13 |
| 20:3(8,11,14) | *cis*-8,11,14-Eicosatrienoic | 0.48 |
| 20:4(5,8,11,14) | Arachidonic | 1.32 |
| 22:0 | Behenic | 1.17 |
| 22:1(9) | Erucic | 1.63 |
| 22:2(13,16) | Docosadienoic | 15.16 |
| Sum | Saturated fatty acids | 36.57 |
| | Unsaturated fatty acids | 63.42 |
| Unsaturation Index | | 1.09673 |

[*]These numbers indicate the percentage of a fatty acid in the total fatty acids detected in onion pollenkitt. For original data on retention time and area, refer to Table S1.

respectively. The detected unsaturated fatty acids comprised 63.42%, which is slightly exceeded the saturated ones at 36.57%.

## Ultrastructure of pollen grain surface with and without pollenkitt

Pollenkitt covers the outer layer of the pollen wall, called exine. High-resolution scanning electron microscopy (SEM) was used to observe the specific morphology of pollen grains covered and uncovered by pollenkite. The surface of native dry pollen grain with pollenkitt appeared less structured in comparison to exine (Fig 3a and b). After removal of pollenkitt using previously developed protocol of pollen preparation for SEM [37], the exine surface can be characterized as finely striated, rugulated and perforated.

With SEM, we were able to visualize the process of pollenkitt being washed away by water solution during sample preparation for study pollenkitt role in pollen tube germination and DNase activity (see sections 2.2.1 and 2.3.1). Fig 3c illustrates the process of pollenkitt removal from pollen grain surface by water solution. To the left of the furrow (sulcus), streams of washed-off lipid-rich pollenkitt are clearly visible; to the right of the furrow, structured exine and washed-off pollenkitt in the form of drops appear. The washed-off pollenkitt remains close to pollen grain (Fig 3d). When the pollenkitt was removed after 10 washing cycles including vortexing, centrifuging, and discarding the supernatant containing pollenkitt, pollen grains were not surrounded by pollenkitt emulsion (Fig 3e).

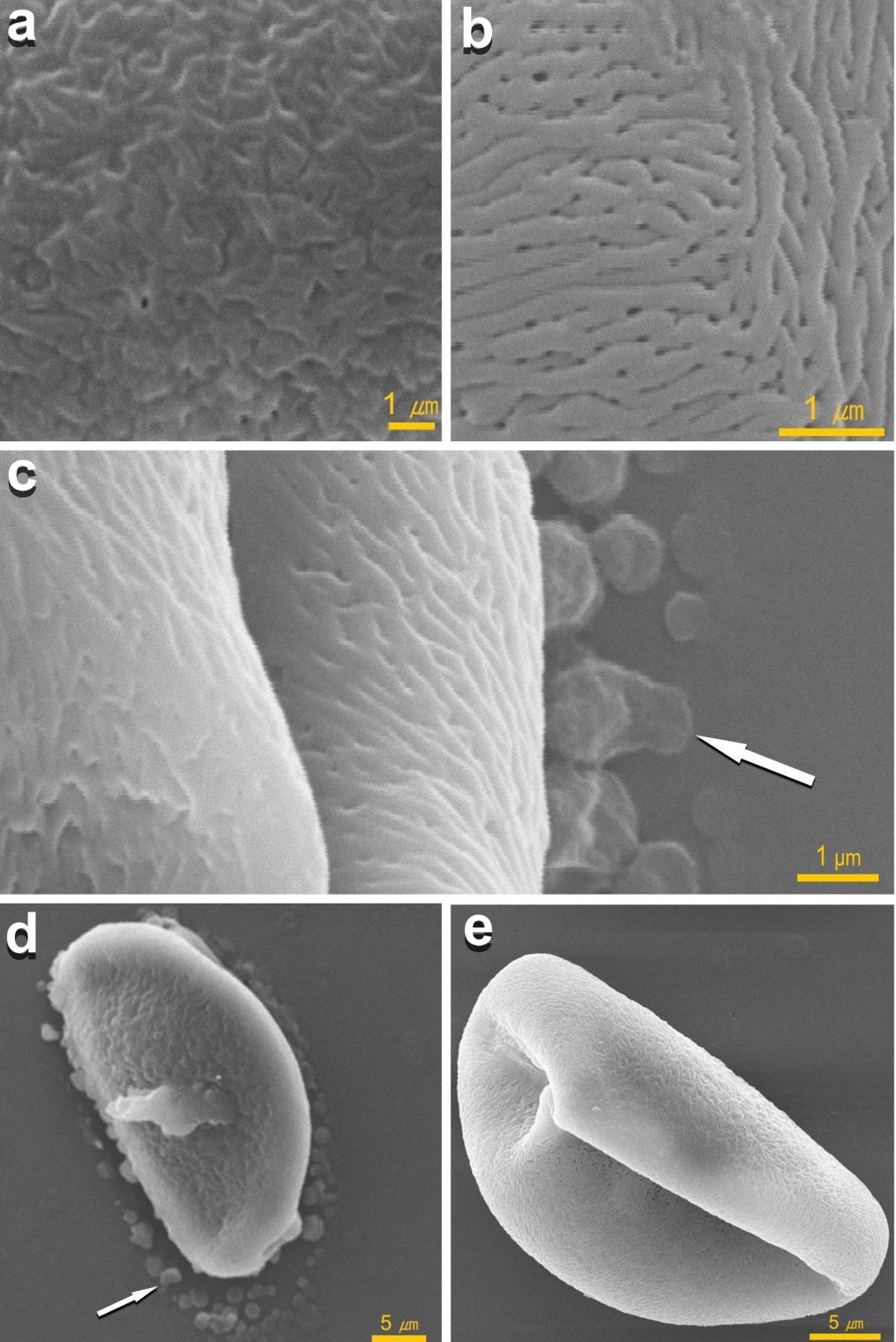

**Fig 3. SEM images of onion pollen grains.** (a) the surface of onion native dry pollen grain with the pollenkitt; (b) the exine surface of onion pollen grain after removal of pollenkitt according to previously developed protocol [33]; (c) removal of pollenkitt from the surface of the pollen grain by washing with an aqueous solution; (d) pollen grain surrounded with emulsion of pollenkitt; (e) pollen grain after removing supernatant containing pollenkitt. Arrows – washed off emulsion of pollenkitt. The images were taken under a scanning electron microscope JEOL JCM-7000.

## Discussion

The pollen grain is a natural carrier of genetic material. It has recently been shown that pollen can be used to deliver biopolymers directly into the zygote [22–27]. Most studies on pollen grains were conducted by palynologists to solve the challenges of plant taxonomy. However, the pollen grain has not been studied in terms of its use as a supervector. Pollenkitt, which covers the exine of the pollen grain, plays a crucial role in pollen functionality. The role of pollenkitt has been highlighted in a number of studies [8,38–40]. In many crops including onion, the role and physiological functions of pollenkitt have not been sufficiently examined yet. Here we present the results of an interdisciplinary approach to the study of onion pollenkitt.

### Pollenkitt is necessary for pollen germination and tube growth on the stigma of onions

Given that genetic manipulations of pollen are carried out in a wet medium, we aimed to elucidate the effects of pollenkitt washing from the pollen surface, pollenkitt absence and mechanical agitation on germination of pollen grain and tube growth within the pistil. Using aniline blue staining and fluorescence microscopy, we showed that when a stigma was pollinated with pollenkitt-free suspension of pollen grains, almost complete inhibition in germination and tube growth was observed. On the other hand, the pollen grains suspension with no removed pollenkitt showed normal germination and growth even after harsh mechanical manipulations like centrifuging and vortexing. The mechanism by which pollenkitt facilitates grains germination and tubes growth is still not well understood. Using SEM, Chichiriccò and colleagues [20] showed that pollenkitt of *Crocus vernus* forms bridges between pollen grains and the stigma. Murphy [41] showed that when the pollen grain comes into contact with the stigma, its pollen coat undergoes reorganization. This process results in the concentration of the pollenkitt at the interface, forming hydrophilic channels likely composed of protein. These channels facilitate the transfer of water and chemical signals from the stigma to the pollen [42]. Our results are consistent with previous findings on the important role of pollenkitt in pollen germination. We showed that the presence of pollenkitt is required for the proper functioning of pollen grains *in vivo*, independent of any mechanical manipulation of pollen and the use of a humid environment for pollination.

### Evidence of DNase activity of onion pollenkitt

In this work, the DNase activity of pollenkitt was established. However, the exact identity of the DNase enzyme(s) responsible remains to be elucidated. The origin of DNase also remains unclear. It has been reported that the tapetum responsible for pollenkitt production releases proteases and nucleases towards the end of its active biosynthetic phase [43–46]. Some studies suggest that DNase-like enzymes are not restricted to the pollenkitt. Broglia and Corona [47] demonstrated that pollen grains release nucleases during incubation in a hypertonic aqueous medium. An exogenous DNA, which was added to the incubation medium, has been degraded within 30 minutes. The authors suggested that the nuclease has diffused from the interior of pollen grains rather than the coating area. In another study [48], the author used cyto-chemical assay to demonstrate the presence of DNase localized in vesicles among the intine region as well as in the region around the generative cell. Our study clearly demonstrated the presence of DNase activity in onion pollenkitt.

### Characteristics of fatty acids in pollenkitt

The study of the fatty acid composition of pollenkitt was carried out on a few plant species, which revealed differences in chemical and quantitative composition [12–17]. Most naturally

occurring fatty acids have an unbranched chain with 4–28 carbon atoms. All fatty acids can be classified as short-chain (1–5 carbon atoms) [49], medium-chain (6–12 carbon atoms), long-chain (13–19 carbon atoms), and very long-chain (>19 carbon atoms) [50].

Chemical analysis of the fatty acid composition of onion pollenkitt established 20 fatty acids including one medium-chain fatty acid (12 carbon atoms), eleven long-chain fatty acids (13–18 carbon atoms) and eight very-long-chain fatty acids (20–22 carbon atoms). Notably, a short-chain fatty acids were not detected in onion pollenkitt. Among species pollenkitt to be chemically characterized only in *Crocus vernus* a small amount (3.56% of the total fatty acids in pollenkitt) of butyric acid (C4) was found [51]. In pollenkitt of lily, which is taxonomically close to onion, short chain fatty acids also have not been detected [21]. All short chain fatty acids are volatile [52] and could serve as a chemical cue for pollinators [53]. However, because of volatility, short chain fatty acids require different approach for probe preparation using adsorbent [53] and could be underrepresented in hexane-washed fatty acids preparations.

The medium chain fatty acids in onion pollenkitt represented by a single type detected in small amount (1.33% of the total fatty acids in pollenkitt) - lauric acid (12:0). For *Brassica napus* pollenkitt, it has been shown that pelargonic acid (9:0) is also been found [41]. When analyzing lily, tobacco and spruce pollenkitt, capric acid (10:0) along with lauric acid, was found [21]. In pollenkitt of clover also a single medium-chain fatty acid, lauric acid, was detected [18] while in pollenkitt of *Crocus vernus* along with lauric acid, capronic (6:0), caprylic (8:0) and caprinic (10:0) fatty acids were detected. In all analyzed species medium-chain fatty acids were minor and accounts for less than 3% of the total fatty acids in pollenkitt. Medium chain fatty acids such as lauric acid, essential for sporopollenin biosynthesis [54,55] and could be presented in pollenkitt as a residual precursor of the exine [51].

The long chain fatty acids are the most abundant and diverse group of fatty acids represented in all analyzed pollenkitts. In onion pollenkitt long chain fatty acids are represents by 11 individual types of both saturated and unsaturated fatty acids. In pollenkitt of all species analyzed to date there are 1–3 major long-chain fatty acids while other fatty acids are presented in less quantities and might be considered as minor. If a single major fatty acid is present in pollenkitt it is always palmitic acid and if the second and the third major fatty acids are presents than at least one of them are unsaturated derivatives of stearic fatty acid [16–19,21,51]. Furthermore, for the entomophilous species like tobacco, lily, sunflower, *Crocus vernus*, clover and *Brassica napus* such pattern can be explained by critical role of the saturated and unsaturated long chain fatty acids for pollinators nutrition [56]. Onion pollenkitt, in contrast to other species, contains a single major long-chain fatty acid – palmitic (18.72% of the total fatty acids in pollenkitt).

The relative content of VLCFAs (very long chain fatty acids) in onion pollenkitt is higher (17.96% of the total fatty acids in pollenkitt) than in lily pollenkitt (11.24% of the total fatty acids in pollenkitt) [21]. A greater variety of VLCFAs types was found in lilies (15 types of fatty acids) compared to onions (3 types of fatty acids). The major VLCFA of onion pollenkitt is docosadienoic acid (15.16% of the total fatty acids in pollenkitt) while other VLCFAs are minor. There are no noticeable major VLCFA in lily (as like in spruce, clover and *Crocus vernus)* pollenkitt, instead, all fatty acids presented in relatively small amount (<4% of the total fatty acids in pollenkitt) [21]. On the contrary, tobacco pollenkitt, also contains the major VLCFA - beheic (29.77% of the total fatty acids in pollenkitt) [21]. In maize pollenkitt from pollen produced by different hybrids a significant quantitative and qualitative difference in VLCFA amount was discovered [19]. Considering a huge amount of pollen produced by plants and absence of data about importance of specific types of VLCFA for pollen hydration, it is possible to assume that different VLCFA are interchangeable which means that probably the total amount of VLCFA is more important for pollen. Therefore, observed difference in VLCFAs composition in different hybrids is a reaction norm.

Onion pollenkitt fatty acids composition quantitatively and qualitatively differ from pollenkitt of other entomophilous species. Observed differences might be explained by species-specific characteristics of metabolism. Nevertheless, technical aspects, like for short-chain fatty acids probe preparation, should be not excluded.

## Conclusions

In conclusion, our study provides important insights into the role of pollenkitt in onion pollen biology, emphasizing its critical functions in pollen germination, DNase activity, and chemical composition. We demonstrated that the removal of pollenkitt significantly hinders pollen tube growth, underscoring its essential role in pollen-stigma interactions. Furthermore, the discovery of DNase activity within onion pollenkitt highlights a potential barrier to gene-editing applications, as exogenous DNA may be rapidly degraded in its presence. The fatty acid analysis revealed a complex composition of 20 fatty acids, predominantly unsaturated. Additionally, SEM imaging provided clear evidence of structural differences between pollen grains with and without pollenkitt, offering further insights into its protective and adhesive properties.

These findings contribute to a deeper understanding of pollenkitt's multi-functionality and emphasize the importance of further research in the future. This knowledge will be crucial for developing more efficient pollen-mediated gene-editing systems, particularly in agriculturally important species like onions. We believe that the findings and observations presented in this work hold significant value for future research. This applies not only to onions but potentially to other species as well.

## Supporting information

**S1 Fig. Onion pollen collected as described in Section 4.1.**
(TIF)

**S2 Fig. The six stages of flower development in onion according to Currah and Ockendon [34]. (1) Opening of flower bud; (2) Start of anther dehiscence; (3) End of anther dehiscence; (4) Appearance of stigmatic knob; (5) Withering of filaments; (6) Start of withering of style.** In our study, pollination was performed during the appearance of the stigmatic knob (stage 4).
(TIF)

**S3 Fig.** chromatogram showing the separation of fatty acid methyl esters The x-axis represents retention time in minutes, while the y-axis indicates the detector response in microvolts (uV).
(TIF)

**S1 Table. The results of GC-FID of onion pollenkitt. retention time (minutes) and peak area (arbitrary unit).**
(PDF)

## Author contributions

**Conceptualization:** Ludmila Khrustaleva.

**Formal analysis:** Majd Mardini, Aleksey Ermolaev, Ludmila Khrustaleva.

**Funding acquisition:** Ludmila Khrustaleva.

**Investigation:** Ludmila Khrustaleva.

**Methodology:** Majd Mardini, Aleksey Ermolaev, Alexey Yu. Kuznetsov.

**Project administration:** Ludmila Khrustaleva.

**Supervision:** Ludmila Khrustaleva.

**Validation:** Sergey Buravkov, Ludmila Khrustaleva.

**Visualization:** Majd Mardini, Ludmila Khrustaleva.

**Writing – original draft:** Majd Mardini, Aleksey Ermolaev, Ludmila Khrustaleva.

**Writing – review & editing:** Majd Mardini, Aleksey Ermolaev, Alexey Yu. Kuznetsov, Alexey V. Zhevnerov, Sergey Buravkov.

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
