## [Decision Letter · Decision Letter 0]

8 Jan 2025

PONE-D-24-54899Onion pollenkitt: Function, DNase activity, fatty acid composition, and SEM imagingPLOS ONE

Dear Dr. Khrustaleva,

Thank you for submitting your manuscript to PLOS ONE. After careful consideration, we feel that it has merit but does not fully meet PLOS ONE’s publication criteria as it currently stands. Therefore, we invite you to submit a revised version of the manuscript that addresses the points raised during the review process.

We look forward to receiving your revised manuscript.

Kind regards,

Muhammad Abdul Rehman Rashid, PhD

Academic Editor

PLOS ONE

Journal requirements: When submitting your revision, we need you to address these additional requirements. 1. Please ensure that your manuscript meets PLOS ONE's style requirements, including those for file naming. The PLOS ONE style templates can be found at https://journals.plos.org/plosone/s/file?id=wjVg/PLOSOne_formatting_sample_main_body.pdf and https://journals.plos.org/plosone/s/file?id=ba62/PLOSOne_formatting_sample_title_authors_affiliations.pdf. 2. PLOS ONE now requires that authors provide the original uncropped and unadjusted images underlying all blot or gel results reported in a submission’s figures or Supporting Information files. This policy and the journal’s other requirements for blot/gel reporting and figure preparation are described in detail at https://journals.plos.org/plosone/s/figures#loc-blot-and-gel-reporting-requirements and https://journals.plos.org/plosone/s/figures#loc-preparing-figures-from-image-files. When you submit your revised manuscript, please ensure that your figures adhere fully to these guidelines and provide the original underlying images for all blot or gel data reported in your submission. See the following link for instructions on providing the original image data: https://journals.plos.org/plosone/s/figures#loc-original-images-for-blots-and-gels.   In your cover letter, please note whether your blot/gel image data are in Supporting Information or posted at a public data repository, provide the repository URL if relevant, and provide specific details as to which raw blot/gel images, if any, are not available. Email us at plosone@plos.org if you have any questions. 3. Thank you for stating the following financial disclosure:  [This research was supported by the Ministry of Science and Higher Education of the Russian Federation, agreement No. 075-152022-317 dated 20 April 2022, “Agrobotechnologies of the Future”. SEM analysis was partly supported by the Russian Foundation for Basic Research, grant number 20-016-00065].  Please state what role the funders took in the study.  If the funders had no role, please state: ""The funders had no role in study design, data collection and analysis, decision to publish, or preparation of the manuscript."" If this statement is not correct you must amend it as needed. Please include this amended Role of Funder statement in your cover letter; we will change the online submission form on your behalf.

Reviewers' comments:

Reviewer's Responses to Questions

**Comments to the Author**

1. Is the manuscript technically sound, and do the data support the conclusions?

Reviewer #1: Yes

Reviewer #2: Partly

Reviewer #3: Yes

2. Has the statistical analysis been performed appropriately and rigorously? 

Reviewer #1: Yes

Reviewer #2: N/A

Reviewer #3: Yes

3. Have the authors made all data underlying the findings in their manuscript fully available?

Reviewer #1: Yes

Reviewer #2: Yes

Reviewer #3: Yes

4. Is the manuscript presented in an intelligible fashion and written in standard English?

Reviewer #1: Yes

Reviewer #2: Yes

Reviewer #3: Yes

5. Review Comments to the Author

Reviewer #1: This paper provides a comprehensive examination of onion (Allium cepa) pollenkitt, focusing on its functional properties, biochemical composition, and structural features. The study is intriguing as it delves into an often-overlooked aspect of plant reproductive biology while employing a combination of biochemical assays and imaging techniques.

Reviewer #2: Manuscript is dealing with interesting topic.

Results are fairly presented.

I suggest that impersonal style should be used through the text.

However, some issues need to be answered:

English must be checked. Typographical errors occur through the text.

All abbreviations used in the text must be explained when first mentioned.

Latin names of plants must be written in Italic.

Lines 65-66: “… fatty acids could be classified as short-chain (1-5 carbon atoms). “

Please correct it, as fatty acids are those with C4 and more carbon atoms.

Lines 72-79 should be deleted. Comparison to other topics is redundant as this is not a review paper.

Lines 92-97: Text starting with: Here, we further explore… should be placed in a separate paragraph as it underlines the aim of the work.

Line 172: dH2O …. Correct it to dH2O. What is d?

Line 172: …ddH2O…. Correct it to ddH2O. What is dd?

Line 201: …the upper hexane layer containing methyl esters.

..methyl esters of what? Be more precise.

Line 215: Jiapeng Chen and Hongbing Liu [32]. Correct to… Chen and Liu [32].

Line 217: In the section SEM analysis of onion pollen grains with and without pollenkitt, authors mentioned Fig. 3 a-2.

Later in Results (lines 239,240, 243…) authors mentioned Fig 1a-c.

And later, in lines 264-270 they mentioned Fig 2a-d, and in line 278 Fig 2f.

This should be corrected. Authors should label figures in order as they appear in the text.

Line 302: * These numbers..

What the asterisk refers to?

Lines 377: short-chain fatty acids (1-5 carbon atoms). See previous comment about classification of fatty acids

Line 408: what is VLCFAs? Explain.

In general discussion needs to be improved.

References should be prepared by the journal instructions.

For the figures: all abbreviations should be explained in the table legend

Reviewer #3: Dear Authors

This study is conducted to determine the various impacts of onion (Allium cepa) pollen pollenkitt on germination, DNase activity, ultrastructure, and fatty acid composition. The most important findings reported include that pollenkitt plays an important role in pollen tube germination and elongation, as expressed by DNase activity and composed of several unsaturated fatty acids, while SEM examined structural variation in pollen grains.

This work represents a major contribution to plant biology and presents new insights into the role and chemical composition of pollenkitt in onion pollen. A holistic approach combining chemical analysis (GC-FID) with structural and functional investigations (SEM) provides deeper insights into the function of pollenkitt. Several sections of the manuscript need clarification, amplification, or correction in order to enhance the overall quality of the manuscript.

The abstract is too long and does not summarize the main points effectively.

Suggestion: The abstract should be shortened while ensuring that the main findings are clearly communicated. Be concise and precise.

Introduction: The introduction is good, but it would be even better if it contained a more thorough literature review, especially regarding the physiological functions of pollenkitt, other than just comparing the results of the keyword search.

The suggestion would be the development of sections by incorporating references that illustrate recent research into the biochemical nature and ecological roles of pollenkitt in a variety of plant species.

Results are presented regularly; however, the format of some figures and tables could be improved for better clarity.

Recommendation: Ensure that all figures and tables have concise captions that describe the data presented, and that they are integrated smoothly into the text.

Some figure legends are not descriptive enough and lack the explanation of abbreviations.

Suggestion: Legends should be revised to include a brief statement of the significance of each image. Also, all abbreviations should be written out in full the first time they are used.

Some methodological steps lack in detail, mainly those related to fatty acids extraction and analysis and DNase activity.

Exhuming: Include more specific protocols, pointing out exactly in which ways they deviated from the standard practice if the protocol was modified; that could be helpful for people in trying to replicate this study further.

While the discussion well summarized the findings, its practical implications of DNase activities, especially in relation to genetic engineering, can further be elaborated on.

Suggestion: Add an explanation of the consequences that these results have when attempting applications, specifically enhancements on PMGE.

The conclusion does summarize the research, but could be stronger.

Suggestion: Highlight the significance of the pollenkitt not only for onions but also its broader applicability in plant biology and gene editing applications.

There is some inconsistency in the formatting of the references.

Suggestion: Check the format of the citations according to the journal instructions and fix any inconsistencies in the author lists, year of publication or

6. PLOS authors have the option to publish the peer review history of their article (what does this mean? ). If published, this will include your full peer review and any attached files.

**Do you want your identity to be public for this peer review?** For information about this choice, including consent withdrawal, please see our Privacy Policy .

Reviewer #1: **Yes: ** Abhishek Appaji

Reviewer #2: No

Reviewer #3: No

---

## [Author Response · Author response to Decision Letter 1]

30 Jan 2025

Response to Reviewer #1:

General comment:

“This paper provides a comprehensive examination of onion (Allium cepa) pollenkitt, focusing on its functional properties, biochemical composition, and structural features. The study is intriguing as it delves into an often-overlooked aspect of plant reproductive biology while employing a combination of biochemical assays and imaging techniques.”

Our response:

Thank you for your careful reading of our manuscript and deep understanding of our idea motivating the study. We appreciate your positive evaluation.

Response to Reviewer #2

General comment:

“Manuscript is dealing with interesting topic. Results are fairly presented. I suggest that impersonal style should be used through the text. However, some issues need to be answered: English must be checked. Typographical errors occur through the text. All abbreviations used in the text must be explained when first mentioned. Latin names of plants must be written in Italic.

In general discussion needs to be improved. References should be prepared by the journal instructions. For the figures: all abbreviations should be explained in the table legend”

Our response:

Thank you for your thoughtful comments and the time you spent reviewing our manuscripts. Regarding your suggestions, we have made the following revisions:

• The manuscript has been checked for English language and typographical errors, and corrections have been made throughout. We also removed all unnecessary hyphenations.

• All abbreviations are fully explained at the first mention.

• The Latin names of plants have been italicized.

• Necessary changes have been made to the references format.

• All abbreviations in the figure legends have been explained as suggested.

We appreciate your valuable comments and believe these changes have improved the manuscript. Below is a step-by-step response to each of the specific comments you provided.

Specific comments:

Reviewer’s remark:

“Lines: 65-66: … fatty acids could be classified as short-chain (1-5 carbon atoms).

Please correct it, as fatty acids are those with C4 and more carbon atoms.”

Our response:

Thank you for your comment. We agreed with you. We have analyzed the literature on this issue. There is no generally accepted classification of fatty acids. This part of the Introduction has been removed and accordingly the following text has been added to the Discussion: “Most naturally occurring fatty acids have an unbranched chain with 4 to 28 carbon atoms” (lines 371-372)

Reviewer’s remark:

“Lines 72-79 should be deleted. Comparison to other topics is redundant as this is not a review paper.”

Our response:

We find it valuable to include this paragraph in the manuscript, particularly in the introduction. By highlighting the scarcity of research on pollenkitt, particularly on agriculturally important species, we draw attention to the need for further research in this area.

Reviewer’s remark:

“Lines 92-97: Text starting with: Here, we further explore… should be placed in a separate paragraph as it underlines the aim of the work.”

Our response:

Thank you for pointing this out. We have revised the manuscript accordingly.

“Line 172: dH2O …. Correct it to dH2O. What is d?”

Reviewer’s remark:

Thank you for your comment. The "d" in dH2O stands for "distilled".

Our response:

We have added an explanation of this abbreviation in the text where it first appears (line 162). Additionally, the "2" in H2O has been correctly formatted as a subscript throughout the manuscript

Reviewer’s remark:

“Line 172: …ddH2O…. Correct it to ddH2O. What is dd?”

Our response:

This is a typo. It should be dH2O. Thank you for pointing this out.

Reviewer’s remark:

“Line 201: …the upper hexane layer containing methyl esters. ..methyl esters of what? Be more precise.”

Our response:

Thank you for pointing this out. We have specified "fatty acid methyl esters". We have made this correction in the text (line 199)

Reviewer’s remark:

“Line 215: Jiapeng Chen and Hongbing Liu [32]. Correct to… Chen and Liu [32].”

Our response:

Changes have been made to the text accordingly (line 213)

Reviewer’s remark:

Line 217: In the section SEM analysis of onion pollen grains with and without pollenkitt, authors mentioned Fig. 3 a-2. Later in Results (lines 239,240, 243…) authors mentioned Fig 1a-c. And later, in lines 264-270 they mentioned Fig 2a-d, and in line 278 Fig 2f.

This should be corrected. Authors should label figures in order as they appear in the text.

Our response:

Thank you for your comment. We have revised the figure references to ensure they are labeled in the proper order as they appear in the text.

Reviewer’s remark:

“Line 302: * These numbers..What the asterisk refers to?”

Our response:

These lines are part of the footnote for Table 1. The asterisk refers to the title of the last column (% in pollenkitt*)

Reviewer’s remark:

“Lines 377: short-chain fatty acids (1-5 carbon atoms). See previous comment about classification of fatty acids”

Our response:

Changes has been made.

Reviewer’s remark:

“Line 408: what is VLCFAs? Explain.”

Our response:

VLCFAs is an abbreviation for very long-chain fatty acids. This has already been explained where it first appears in the text (line 410).

Response to Reviewer #3

General comment from the reviewer:

“This work represents a major contribution to plant biology and presents new insights into the role and chemical composition of pollenkitt in onion pollen. A holistic approach combining chemical analysis (GC-FID) with structural and functional investigations (SEM) provides deeper insights into the function of pollenkitt. Several sections of the manuscript need clarification, amplification, or correction in order to enhance the overall quality of the manuscript.”

Our response:

Thank you for the time and effort spent revising our manuscript. We have carefully reviewed all suggestions and made the necessary revisions to improve the manuscript's quality and clarity. We have addressed each of the specific comments in the table below.

Specific comments:

Reviewer’s remark:

“The abstract is too long and does not summarize the main points effectively.

Suggestion: The abstract should be shortened while ensuring that the main findings are clearly communicated. Be concise and precise.”

Our response:

We believe the abstract properly reflects our findings while being concise. Following PLOS ONE's guidelines, which suggest a 300-word limit, our abstract is 175 words.

Reviewer’s remark:

“Introduction: The introduction is good, but it would be even better if it contained a more thorough literature review, especially regarding the physiological functions of pollenkitt, other than just comparing the results of the keyword search. The suggestion would be the development of sections by incorporating references that illustrate recent research into the biochemical nature and ecological roles of pollenkitt in a variety of plant species.”

Our response:

The physiological functions of pollenkitt are thoroughly covered in lines 49-57 of the introduction, with sufficient supporting literature (references [6-11]). We believe this section provides a comprehensive overview of the topic in the context of our study.

Reviewer’s remark:

“Results are presented regularly; however, the format of some figures and tables could be improved for better clarity.

Recommendation: Ensure that all figures and tables have concise captions that describe the data presented, and that they are integrated smoothly into the text.”

Our response:

We reviewed the entire manuscript and made the necessary changes and corrections in the journal’s format to ensure clarity.

Reviewer’s remark:

“Some figure legends are not descriptive enough and lack the explanation of abbreviations.

Suggestion: Legends should be revised to include a brief statement of the significance of each image. Also, all abbreviations should be written out in full the first time they are used.”

Our response:

We have formatted all figure captions according to the PLOS ONE guidelines. Each caption should be consisted of two main parts: the figure title and the legend. We have ensured that all captions in our manuscript are laid out accordingly. There is no requirement for a "statement of the significance" of the image in the caption (please refer to this link: https://journals.plos.org/plosone/s/file?id=wjVg/PLOSOne_formatting_sample_main_body.pdf")

Nonetheless, all scientific aspects of the figures are thoroughly presented either in the Results or the Discussion sections. Regarding the abbreviations, we have reviewed the manuscript and ensured that all abbreviations are written out in full the first time they are used for clarity.

Reviewer’s remark:

“Some methodological steps lack in detail, mainly those related to fatty acids extraction and analysis and DNase activity.

Exhuming: Include more specific protocols, pointing out exactly in which ways they deviated from the standard practice if the protocol was modified; that could be helpful for people in trying to replicate this study further.”

Our response:

We believe that all technical procedures are presented in sufficient detail to ensure reproducibility. Please refer to the Materials and Methods section, particularly lines 163-192 (on DNase activity analysis) and lines 196-216 (on fatty acid extraction). Additionally, a comprehensive protocol for the SEM experiments is thoroughly presented in our earlier paper (Ermolaev et al., 2024), which is also properly cited in the manuscript [37].

Reviewer’s remark:

“While the discussion well summarized the findings, its practical implications of DNase activities, especially in relation to genetic engineering, can further be elaborated on.

Suggestion: Add an explanation of the consequences that these results have when attempting applications, specifically enhancements on PMGE.”

Our response:

This issue is clearly addressed in many places of the manuscript (lines 336-354) and also in the Conclusion section.

Reviewer’s remark:

“The conclusion does summarize the research, but could be stronger

Suggestion: Highlight the significance of the pollenkitt not only for onions but also its broader applicability in plant biology and gene editing applications.”

Our response:

See the previous comment.

Reviewer’s remark:

“There is some inconsistency in the formatting of the references.

Suggestion: Check the format of the citations according to the journal instructions and fix any inconsistencies in the author lists, year of publication or”

Our response:

Thank you for bringing this to our attention. We have made all necessary changes to the manuscript accordingly.

---

## [Decision Letter · Decision Letter 1]

3 Mar 2025

Onion pollenkitt: Function, DNase activity, fatty acid composition, and SEM imaging

PONE-D-24-54899R1

Dear Dr. Khrustaleva,

We’re pleased to inform you that your manuscript has been judged scientifically suitable for publication and will be formally accepted for publication once it meets all outstanding technical requirements.

Kind regards,

Muhammad Abdul Rehman Rashid, PhD

Academic Editor

PLOS ONE

Reviewers' comments:

Reviewer's Responses to Questions

**Comments to the Author**

1. If the authors have adequately addressed your comments raised in a previous round of review and you feel that this manuscript is now acceptable for publication, you may indicate that here to bypass the “Comments to the Author” section, enter your conflict of interest statement in the “Confidential to Editor” section, and submit your "Accept" recommendation.

Reviewer #1: All comments have been addressed

Reviewer #2: All comments have been addressed

2. Is the manuscript technically sound, and do the data support the conclusions?

Reviewer #1: Yes

Reviewer #2: Yes

3. Has the statistical analysis been performed appropriately and rigorously? 

Reviewer #1: Yes

Reviewer #2: Yes

4. Have the authors made all data underlying the findings in their manuscript fully available?

Reviewer #1: Yes

Reviewer #2: Yes

5. Is the manuscript presented in an intelligible fashion and written in standard English?

Reviewer #1: Yes

Reviewer #2: Yes

6. Review Comments to the Author

Reviewer #1: No comments

All inputs and suggestion given is met by the authors.

There are no concerns regarding the paper.

Reviewer #2: Manuscript has been improved according to the comments and suggestions given by the reviewers. I have no further comments

7. PLOS authors have the option to publish the peer review history of their article (what does this mean? ). If published, this will include your full peer review and any attached files.

**Do you want your identity to be public for this peer review?** For information about this choice, including consent withdrawal, please see our Privacy Policy .

Reviewer #1: **Yes: ** Abhishek Appaji

Reviewer #2: No

---

## [Editor Report · Acceptance letter]

PONE-D-24-54899R1

PLOS ONE

Dear Dr. Khrustaleva,

I'm pleased to inform you that your manuscript has been deemed suitable for publication in PLOS ONE. Congratulations! Your manuscript is now being handed over to our production team.

Kind regards,

on behalf of

Dr. Muhammad Abdul Rehman Rashid

Academic Editor

PLOS ONE